# Dust Dry Deposition over Israel

**Pavel Kishcha** [1,*] , **Evgeni Volpov** [2] , **Boris Starobinets** [1] , **Pinhas Alpert** [1] **and Slobodan Nickovic** [3]

1   Department of Geophysics, Tel Aviv University, Tel Aviv 69978, Israel; bstarobin@gmail.com (B.S.); pinhas@tauex.tau.ac.il (P.A.)
2   Israel Electric Corporation Ltd., Haifa 3100001, Israel; evgeni.volpov@gmail.com
3   Republic Hydrometeorological Service of Serbia, 11030 Belgrade, Serbia; nickovic@gmail.com
*   Correspondence: pavel@cyclone.tau.ac.il

**Abstract:** Similar quasiperiodic year-to-year variations of dust dry deposition (DDD) with a two–three-year period were found over Israel and north-east Africa. This phenomenon of quasiperiodic interannual variations of DDD has not been discussed in previous publications. Moreover, similar seasonal variations of DDD were found over both Israel and north-east Africa, characterized by significant dust deposition in spring and a decrease in DDD from spring to autumn. These findings indicate the existence of the same causal factors for interannual and seasonal variations of DDD over the two regions, such as similar surface winds created by Mediterranean cyclones. Daily runs of the Dust REgional Atmospheric Model (DREAM) at Tel Aviv University from 2006 to 2019 were used to investigate the main features of the spatio-temporal distribution of dust dry deposition in the eastern Mediterranean, with a focus on Israel. DREAM showed that, on average, during the 14-year study period, in the winter, spring, and summer months, the spatial distribution of monthly-accumulated DDD over Israel was non-uniform with the maximum of DDD over southern Israel. In the autumn months, DREAM showed an increase in DDD over northern Israel, resulting in an almost uniform DDD pattern. The knowledge of DDD spatio-temporal distribution is helpful for understanding the negative effects of DDD on the performance of solar panels and on insulator flashover in the Israel power electric network.

**Keywords:** desert dust; dust dry deposition; quasiperiodic interannual variations; regional modeling; eastern Mediterranean

## 1. Introduction

Satellite and ground-based measurements show that the eastern Mediterranean is a crossroad for various species of air pollution [1]. These species include desert dust from the Sahara Desert as well as from deserts in the Middle East, anthropogenic aerosols from Europe, and marine sea-salt aerosols generated by winds at the Mediterranean Sea surface. Large amounts of air pollutants create a severely polluted environment over coastal areas of the eastern Mediterranean.

Desert dust is the major pollutant over Israel and surrounding areas, influencing air quality and visibility [2–4]. Semi-arid and desert regions, characterized by low cloudiness, are the most attractive locations for solar energy production. Dust deposition on solar panels deteriorates the performance of such panels leading to significant losses in the generated power [5,6]. In addition, dust dry deposition on insulators could cause insulator flashover leading to electricity outages. Such insulator flashovers in the Israel power electric network are discussed by Volpov and Kishcha [7]. Because of the above-mentioned dust-related problems, the investigation of interannual and seasonal variations, as well as spatial distribution of dust dry deposition, is important. Dust dry deposition also

has beneficial effects on soil properties as it is a source of nutrients, such as potassium, phosphorous, iron, sulfur, ammonia, and organic matter [8].

However, little is known about the specific features of spatio-temporal distribution of dust dry deposition (DDD) over Israel and surrounding areas. Ganor and Foner [9] measured dust dry deposition over Israel and the Sinai Desert, from 1970 to 1973. They found significant dust dry deposition over the Negev Desert, located in southern Israel [9]. Dust modeling and forecasts by the Tel-Aviv University (TAU) dust prediction system [10–12] have been carried out from 2006 to 2019. This provides us with the opportunity to estimate spatio-temporal distribution of dust dry deposition over the model domain. Our study was aimed at investigating spatial patterns of annually-, seasonally-, and monthly-accumulated dust dry deposition over the study region, including Israel and surrounding areas (Figure 1C). In addition, we investigated long-term trends in dust dry deposition during the 14-year study period (2006–2019).

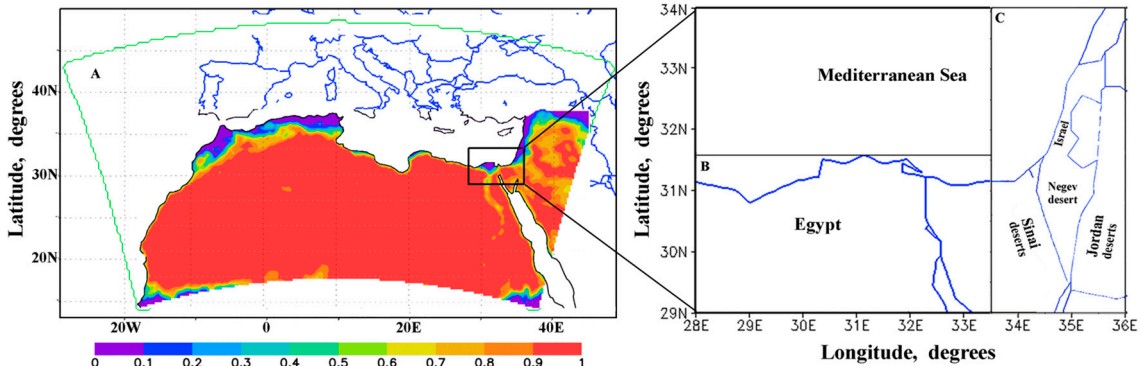

**Figure 1.** Maps of (**A**) the whole Dust REgional Atmospheric Model (DREAM) domain (15° N–50° N; 20° W–45° E) including (**B**) north-east Africa (29° N–31.5° N; 28° E–33.5° E) and (**C**) the study region (Israel and surrounding areas) (29° N–34° N; 33.5° E–36° E). The colors designate the distribution of the potential dust source parameter $\alpha$, which is specified by the global vegetation data [13]. $\alpha$ ranges from 0 (no dust emissions) to 1.

This knowledge will advance the general understanding of the above-mentioned dust-related problems in the south-eastern Mediterranean and in Israel in particular.

## 2. Data and Methodology

Daily model runs of the TAU aerosol prediction system from 2006 to 2019 provide us with consistent model data of dust dry deposition during the 14-year study period. The TAU dust prediction system is based on the Dust Regional Atmospheric Model (DREAM) [13]. A brief outline of the DREAM model is given in the following section.

Our focus was on investigating characteristic features of spatio-temporal distribution of DDD over the study region. As known, Saharan dust intrusions are a common phenomenon over Israel [14]. During these intrusions, dust is transported across north-east Africa, located in proximity to the region under study. In order to understand the relationship between the dust over North Africa and that over the study region, we made a comparison of year-to-year variations, as well as seasonal variations of DDD between north-east Africa and Israel. Accordingly, we made a comparison of year-to-year variations as well as seasonal variations of surface wind speed between north-east Africa and Israel. This was carried out using surface wind M2IMNXLFO [15] at the surface layer height (approximately 60 m) from the NASA MERRA-2 reanalysis including data assimilation [16]. In the process of data assimilation, observations, collected from all available sources, were combined with forecast output from a weather prediction model. The resulting reanalysis is considered to be the 'best' estimate of the state of the atmosphere at a particular time [17].

### 2.1. The DREAM Model

As illustrated by the flowchart (Figure 2), the DREAM model simulates all major processes of the atmospheric dust cycle, such as dust generation, transport, and dry and wet deposition [13]. The NCEP/Eta (National Centre of Environmental Prediction) regional meteorological model is the driver for the DREAM dust model [13]. At the preprocessing step (Figure 2), the NCEP/Eta model is initialized with the NCEP analysis, and the lateral boundary data are updated every six hours by the Global Forecast System (GFS) model of NCEP. The DREAM dust module (Figure 2) calculates a 3D distribution of dust pollution over the model domain (20° W–45° E; 15° N–50° N) with 0.3° horizontal grid spacing and 24 vertical levels up to 15 km in height (Figure 1A). Eight particle size classes are used in the model with the particle size (diameter) between 0.2–0.36; 0.36–0.6; 0.6–1.2; 1.2–2; 2–3.6; 3.6–6; 6–12; 12–20 μm [18]. The dust module is initialized with 3D dust distributions obtained from previous dust module runs (from previous days). DREAM includes dust sources located in the western, central, and eastern Sahara, as well as in the Arabian Peninsula (Figure 1A). This is based on the United States Geological Survey (USGS) data of vegetation cover [13]. Dust emissions depend on soil moisture, soil texture, friction velocity, and surface conditions [13]. DREAM calculates wind friction velocity based on surface roughness and soil moisture. The DREAM dry deposition scheme follows Giorgi [19]. To calculate dry deposition velocity, this scheme takes into account processes of turbulent diffusion, Brownian diffusion, gravitational settling, and processes of interception and impaction on surface roughness elements [13,19]. A possible effect of sea-breeze on dust dry deposition over Israel was not investigated. This is because a model of higher horizontal resolution than DREAM is needed.

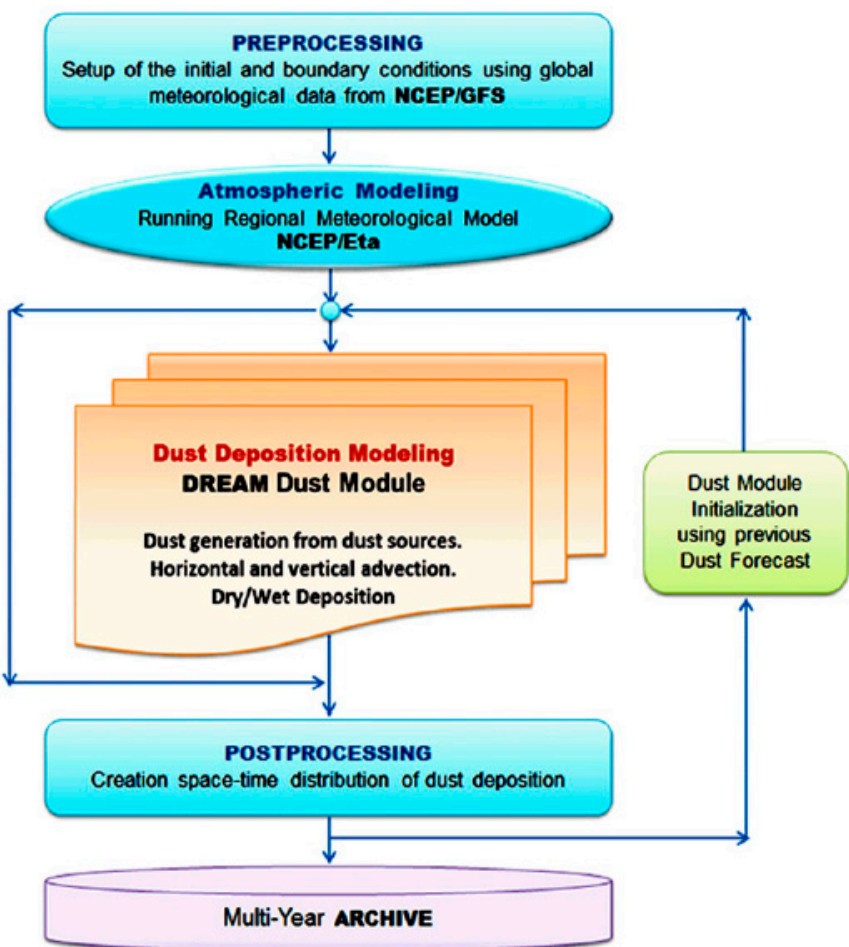

**Figure 2.** Flowchart of the Tel-Aviv University (TAU)/DREAM dust model system used for daily operational dust forecasts over the Mediterranean region from 2006 to 2019.

### 2.2. Validation of Modeled Monthly-Accumulated Dust Dry Deposition

Daily runs of the DREAM dust prediction system, from 2006 to 2019, were used in the current study to obtain spatial distributions of monthly-accumulated dust dry deposition over Israel and surrounding areas, averaged over the 14-year study period. Monthly-accumulated dust deposition model data were validated by Volpov and Kishcha [7] using a comparison with available direct measurements of total air pollution deposited on electric insulators, in the Israel power supply network. This comparison was carried out for different geographic locations and various exposure periods. A good correspondence was obtained between modeled dust deposition data and measurements, characterized by the correlation coefficient of 0.8 ([7], their Figure 9).

In the current study, spatial distribution of modeled annually-accumulated dust deposition, averaged over the 14-year study period, was compared with that based on the only available measurements of dust dry deposition over Israel and the Sinai Peninsula [9]. The latter measurements were conducted during the period from 1970 to 1973 by Ganor and Foner [9]. They measured deposition produced by total suspended dust particles of size (diameter) up to 100 μm. However, DREAM only took into account dust particles of size less than 20 μm. For the purpose of verifying spatial distribution of model data, we used annually-accumulated dust deposition values normalized on dust deposition in Sede-Boker (30.855° N, 34.782° E), located in the middle of the Negev Desert. (This comparison is discussed in Section 3.2).

## 3. Results

We investigated specific features of spatio-temporal distribution of monthly-accumulated, seasonally-accumulated, and annually-accumulated dust dry deposition over the region under study, based on daily runs of the DREAM dust prediction system from 2006 to 2019.

### 3.1. Year-to-Year Variations of Annually-Accumulated Dust Deposition over Israel

We analyzed year-to-year variations of DREAM-based annually-accumulated dust dry deposition (DDD) over specified sites located in the northern, central, and southern parts of the study region (Figure 3a). One can see noticeable quasiperiodic year-to-year variations with a 2–3 year period, which are highly correlated over all parts of the study region (Table 1).

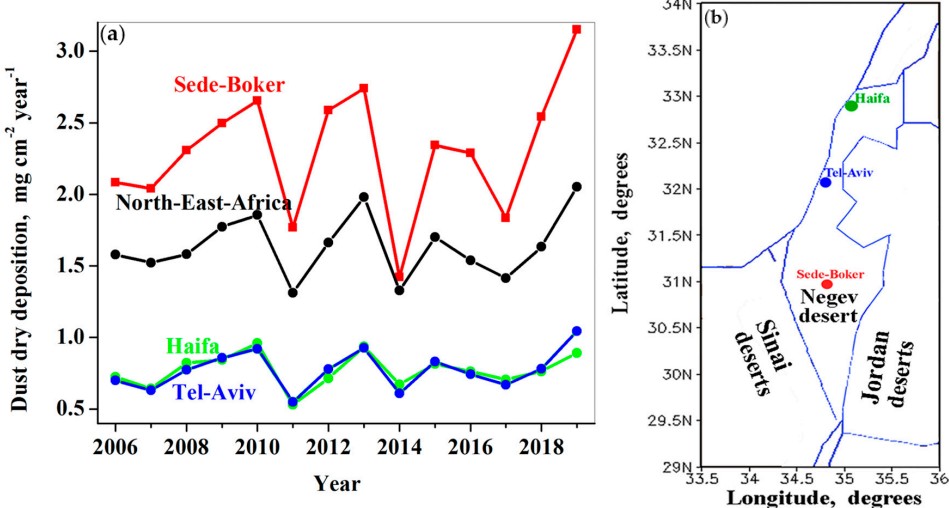

**Figure 3.** (**a**) Year-to-year variations of DREAM-based annually-accumulated dust dry deposition at the specified sites, located in the northern, central, and southern parts of Israel, and over north-east Africa (29° N–31.5° N; 28° E–33.5° E), located in close proximity to the region under study. (**b**) Geographic locations of the specified sites: Haifa (32.79° N; 34.99° E), Tel-Aviv (32.085° N; 34.781° E), and Sede-Boker (30.855° N, 34.782° E).

**Table 1.** Correlation matrix for year-to-year variations of annually-accumulated dust dry deposition (DDD) in Haifa, Tel-Aviv, Sede-Boker, and over north-east Africa during the study period (2006–2019). The numbers represent correlation coefficients (R) together with their standard errors [20]. These correlation coefficients are statistically significant at the 95% confidence level (*p*-values ≤ 0.05).

|  | Haifa | Tel-Aviv | Sede-Boker | North-East Africa |
|---|---|---|---|---|
| Haifa | 1.00 | 0.92 ± 0.10 | 0.77 ± 0.20 | 0.87 ± 0.10 |
| Tel-Aviv | 0.92 ± 0.10 | 1.00 | 0.92 ± 0.10 | 0.97 ± 0.10 |
| Sede-Boker | 0.77 ± 0.20 | 0.92 ± 0.10 | 1.00 | 0.94 ± 0.10 |
| north-east Africa | 0.87 ± 0.10 | 0.97 ± 0.10 | 0.94 ± 0.10 | 1.00 |

As mentioned in Section 2.2, we also analyzed year-to-year variations of DREAM-based annually-accumulated DDD over north-east Africa (Figure 1B), located in proximity to the region under study. DREAM showed a strong correlation between the year-to-year variations over north-east Africa and those at the specified sites (Figure 3 and Table 1). This indicates that the quasiperiodic year-to-year variations of annually-accumulated dust dry deposition over both Israel and north-east Africa were caused by the same factors. To specify these factors, we compared the year-to-year variations of surface wind speed averaged over north-east Africa with those averaged over the southern part of the study region (Figure 4). As mentioned in Section 2, this was carried out using surface wind at the surface layer height (approximately 60 m) from the NASA MERRA-2 reanalysis including data assimilation. One can see similar quasiperiodic year-to-year variations of surface wind speed over the two regions (Figure 4). These quasiperiodic interannual variations of surface wind speed are highly correlated (with the statistically significant correlation coefficient of 0.84 ± 0.15), especially in the last decade. Moreover, the quasiperiodic interannual variations of surface wind speed over the two regions correlated with quasiperiodic interannual variations of DDD over those regions (Figures 3 and 4).

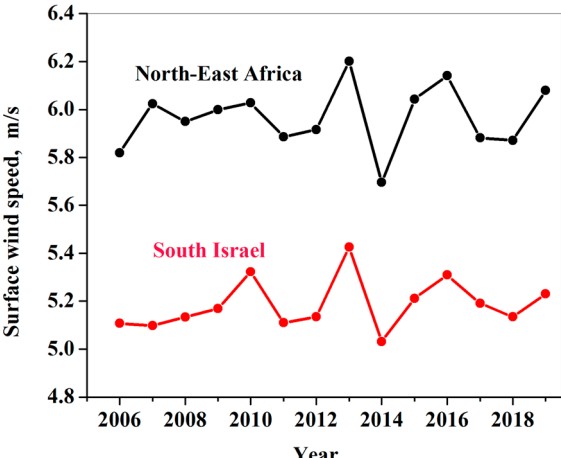

**Figure 4.** Year-to-year variations of the 14-year mean surface wind speed over the southern part of the study region (29.5° N–31.5° N; 34.5° E–35.5° E) and north-east Africa (29° N–31.5° N; 28° E–33.5° E), based on NASA MERRA reanalysis (2006–2019) [15].

As known, dust occurrence in the south-east Mediterranean is associated with cyclonic activity all over the Mediterranean basin [4,14]. These Mediterranean cyclones determine surface winds over the whole area of the south-eastern Mediterranean (including north-east Africa and Israel), causing similar interannual variability of DDD.

In spite of high correlation in the year-to-year variations of DDD over different parts of Israel, there was an essential difference in dust deposition values. In different years, DDD over southern Israel (Sede-Boker) was approximately three times higher than over northern Israel (Haifa) (Figure 3). This will be discussed in the next Section 3.2.

*3.2. Spatial Non-Uniformity of Annually-Accumulated Dust Dry Deposition over Israel*

Taking into account that Israel is distant from major dust sources in the Sahara, one could expect a homogeneous distribution of DDD (assuming that desert dust over Israel was transported mainly from remote sources). However, DREAM showed that this is not the case. This is because the spatial distribution of annually accumulated dust dry deposition (averaged over the 14-year study period) was non-uniform (Figure 5). With respect to the south–north direction, dust dry deposition was essentially higher over the southern part of Israel than over the central and northern parts: DDD in Sede-Boker (southern Israel) was three times higher than in Haifa (northern Israel). This indicates the contribution of dust from the Negev Desert and from adjacent deserts in Jordan and Sinai to the total dust deposition over Israel.

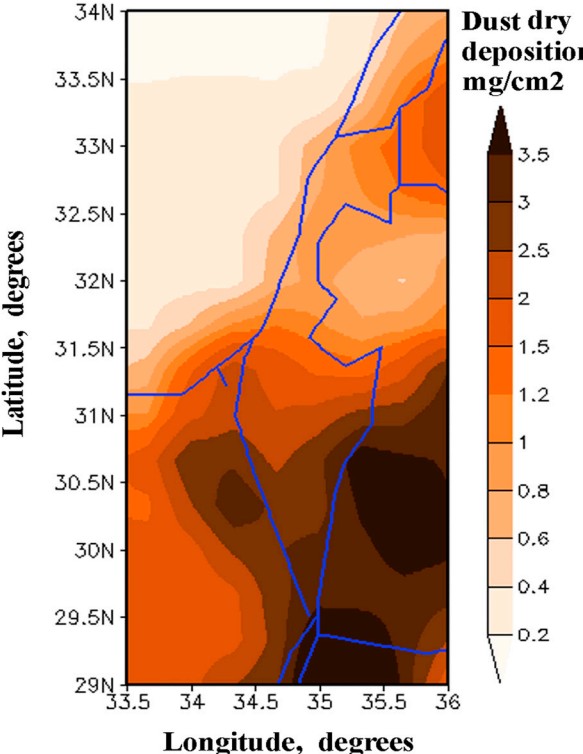

**Figure 5.** Spatial distribution of annually-accumulated dust dry deposition over Israel and surrounding areas, averaged over the 14-year study period.

Our findings about spatial non-uniformity of dust dry deposition are in line with Ganor and Foner [9]. We compared the obtained spatial distribution of 14-year mean modeled annually-accumulated dust deposition with that of measured annually-accumulated dust deposition (Figure 6a) [9]. As mentioned in Section 2.2, the modeled and measured annually-accumulated dust deposition were normalized on the annually-accumulated dust deposition in Sede-Boker (30.855° N, 34.782 °E), located in the middle of the Negev Desert. Figure 6b represents a comparison between modeled and measured annually-accumulated dust dry deposition at the specified sites, located in the south-north direction along Israel. One can see a good correspondence between DREAM dust deposition data and measurements, which supports our findings based on DREAM model data. Both measurements and model data showed DDD values significantly decreasing with distance from the Negev Desert (Sede-Boker) (Figure 6a,b).

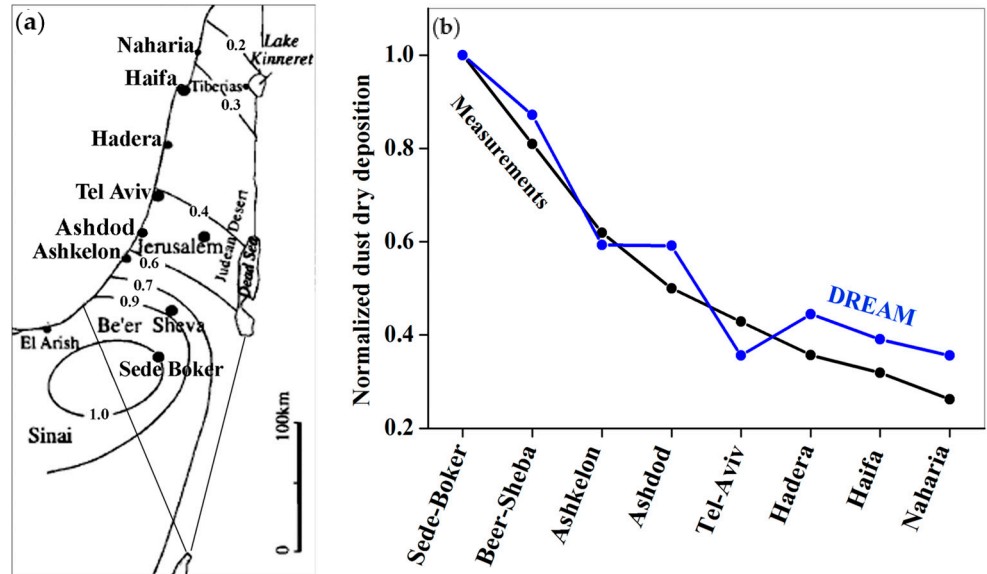

**Figure 6.** (**a**) Spatial distribution of measured annually-accumulated dust dry deposition over Israel and surrounding areas (adapted from Ganor and Foner [9]). The measured annually-accumulated dust dry deposition data were normalized on the annually-accumulated dust deposition in Sede-Boker (30.855° N, 34.782° E). (**b**) Comparison between modeled and measured annually-accumulated dust dry deposition at the specified sites, located in the south–north direction along Israel: Sede-Boker (30.855° N, 34.782° E), Beer-Sheba (31.25° N; 34.80° E), Ashkelon (31.639° N; 34.521° E), Ashdod (31.834° N; 34.637° E), Tel-Aviv (32.085° N; 34.781° E), Hadera (32.47° N; 34.881° E), Haifa (32.79° N; 34.99° E), and Naharia (33.002° N; 35.091° E).

### 3.3. Seasonal Variations of Monthly-Accumulated Dust Dry Deposition over Israel

In order to further investigate the spatial non-uniformity of dust dry deposition, we analyzed the spatial distribution of seasonally-accumulated DDD (Figure 7). DREAM showed a noticeable difference in the patterns of seasonally-accumulated DDD in various seasons. The model DDD spatial distribution in winter, spring, and summer was similar to the distribution of annually-accumulated DDD, characterized by the maximum of DDD over southern Israel (Figure 7a–c). This allowed us to assume that local dust from the Negev Desert contributed to modeled DDD over Israel in winter, spring, and summer. By contrast, in autumn, the spatial distribution of modeled DDD, over the study region, was more uniform than in other seasons (Figure 7d). As discussed in Section 3.4, we assume that the arrival of dust from Syria to northern and central Israel led to this almost uniform spatial distribution of DDD in Israel in autumn.

DREAM showed that, on average, during our study period, there were strong seasonal variations of dry dust deposition with significant DDD in spring (March) and a decrease in DDD from spring to autumn (Figure 8a,b). We also analyzed seasonal variations of DREAM-based monthly-accumulated DDD over north-east Africa (29° N–31.5° N; 28° E–33.5° E) (Figure 9). Our analysis showed similar DDD seasonal variations over both north-east Africa and Israel, characterized by significant dust deposition in spring and a decrease in DDD from spring to autumn. This similarity indicates the same causal factors responsible for the seasonal variations of DDD over Israel and over north-east Africa.

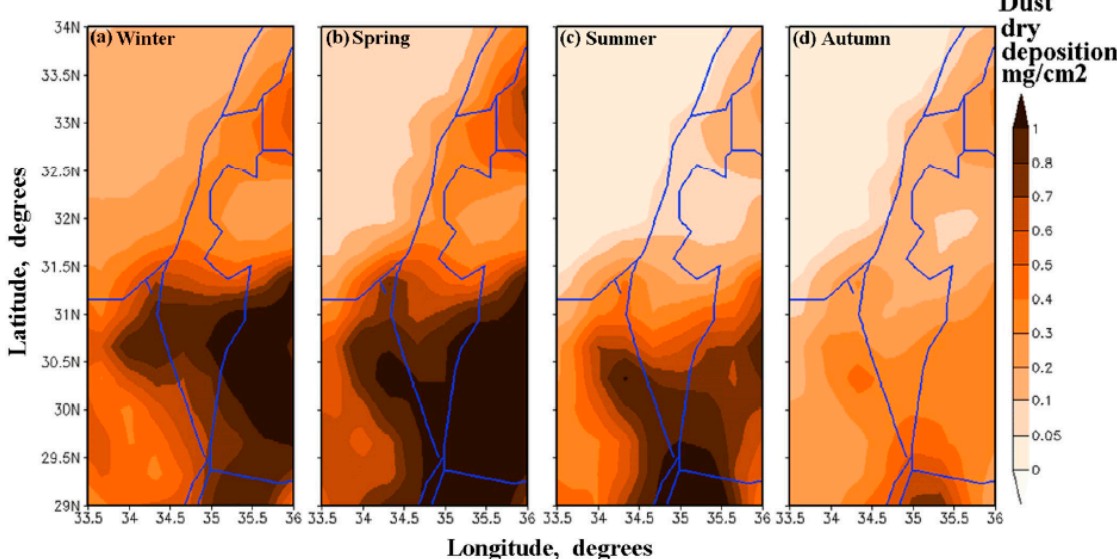

**Figure 7.** Spatial distribution of seasonally-accumulated dust dry deposition over Israel and surrounding areas in (a) winter, (b) spring, (c) summer, and (d) autumn, averaged over the 14-year study period.

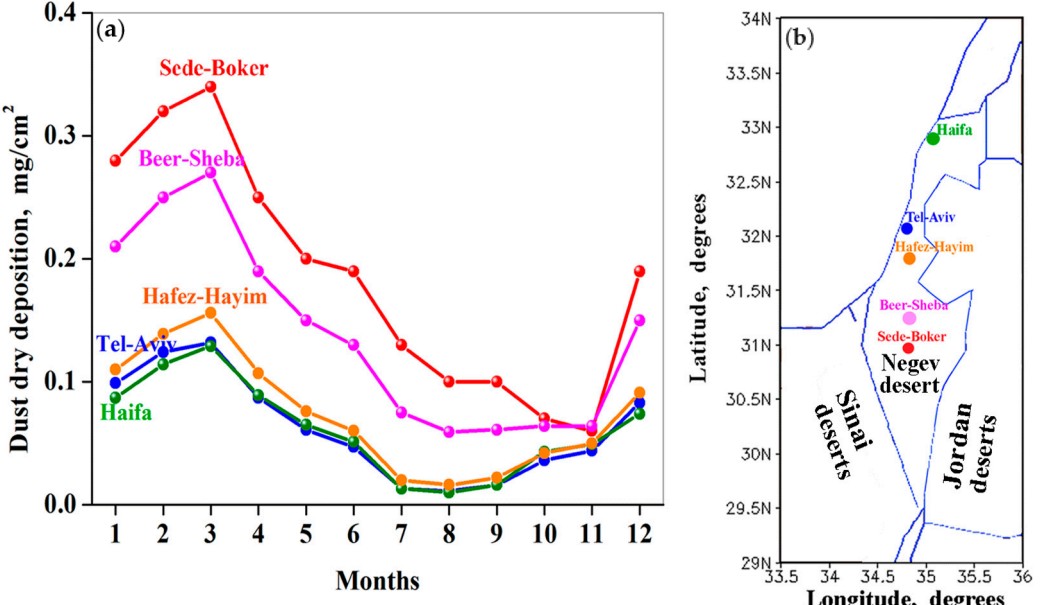

**Figure 8.** (**a**) Fourteen-year mean seasonal variations of DREAM dust dry deposition at the specified sites located in the southern, central, and northern parts of Israel. (**b**) Geographic locations of the specified sites: Haifa (32.79° N; 34.99° E), Tel-Aviv (32.085° N; 34.781° E), Hafez-Hayim (31.79° N; 34.81° E), Beer-Sheba (31.25° N; 34.80° E), and Sede-Boker (30.855° N, 34.782° E).

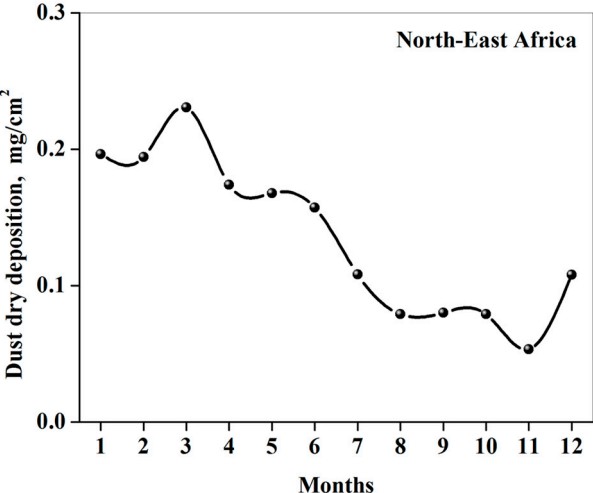

**Figure 9.** Fourteen-year mean seasonal variations of DREAM dust dry deposition over north-east Africa (29° N–31.5° N; 28° E–33.5° E).

We found that the above mentioned similar seasonal variations of DDD were associated with similar seasonal variations of 14-year mean surface wind speed. Figure 10 represents seasonal variations of the 14-year mean surface wind speed, averaged over north-east Africa, and that averaged over the southern part of the study region. One can see that surface wind speed in autumn was lower than in other seasons (Figure 10). As mentioned, Mediterranean cyclones determine surface winds over the whole area of the south-east Mediterranean, including north-east Africa and Israel. The obtained similar seasonal variations of wind speed prove this fact. The similar seasonal variations of DDD and surface wind indicate the presence of a causal link between them.

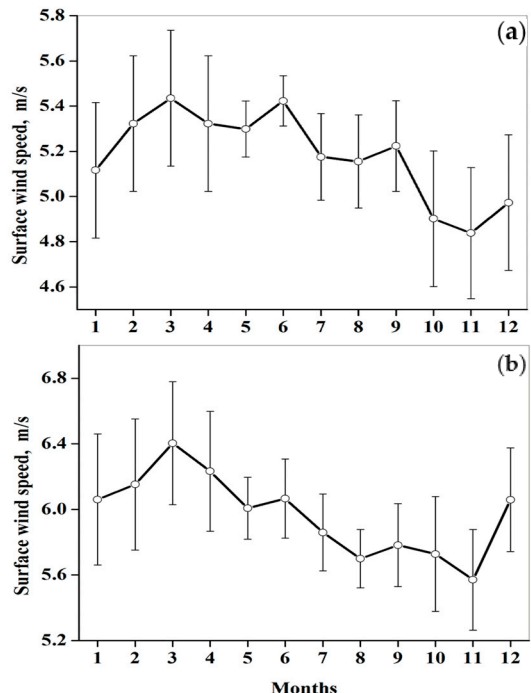

**Figure 10.** Month-to-month variations of the 14-year mean surface wind speed over (**a**) the southern part of the study region (29.5° N–31.5° N; 34.5° E–35.5° E) and (**b**) north-east Africa (29° N–31.5° N; 28° E–33.5° E), based on NASA MERRA reanalysis (2006–2019) [15,16]. The vertical lines designate standard deviation.

There was an essential difference in DDD values between the southern part, on the one hand, and the central and northern parts, on the other hand. In particular, over the southern part, dust deposition noticeably decreased from Sede-Boker through Beer-Sheba to Hafez-Hayim, along a distance of ~100 km (Figure 8). By contrast, along the same distance of ~100 km from Tel Aviv to Haifa, there was almost no difference in dust deposition: seasonal variations of dust dry deposition in Tel Aviv and Haifa coincided (Figure 8). Our findings imply an important point that local dust from the Negev Desert and adjacent deserts in Sinai and Jordan significantly contributed to total dust deposition over southern Israel, causing pronounced non-uniformity of DDD over the study region.

Another factor should be taken into account, in order to understand seasonal variations of DDD over northern Israel, where there is no local dust. Over the northern and central Israel (where Tel-Aviv and Haifa are located), the seasonal minimum of DDD was observed in July–August (Figure 8). This was in contrast to southern Israel, where the seasonal minimum of DDD was observed in October–November (Figure 8). As DREAM shows, over northern and central Israel, dust dry deposition increases from September to November: it can be assumed some dust penetration from Syria (Figure 11a–c). At the same time, from September to November, over southern Israel, a decrease in dust dry deposition takes place (Figure 11a–c). This explains the different seasonal minimum of DDD over northern and southern Israel.

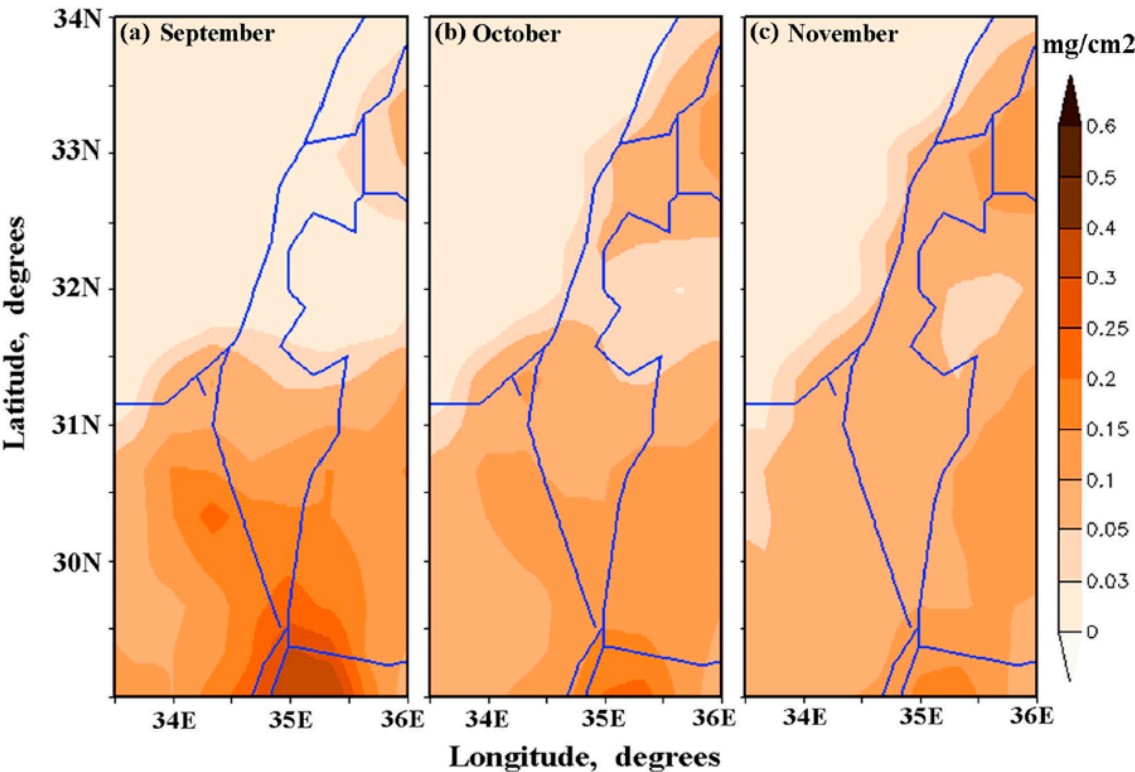

**Figure 11.** Maps of 14-year mean monthly-accumulated dust dry deposition in (**a**) September, (**b**) October, and (**c**) November.

Moreover, the above-mentioned increase in DDD over northern Israel and the decrease over southern Israel in November resulted in the almost uniform distribution of DDD over Israel. Although, the dust deposition values were noticeably lower in November than in other seasons (Figure 12).

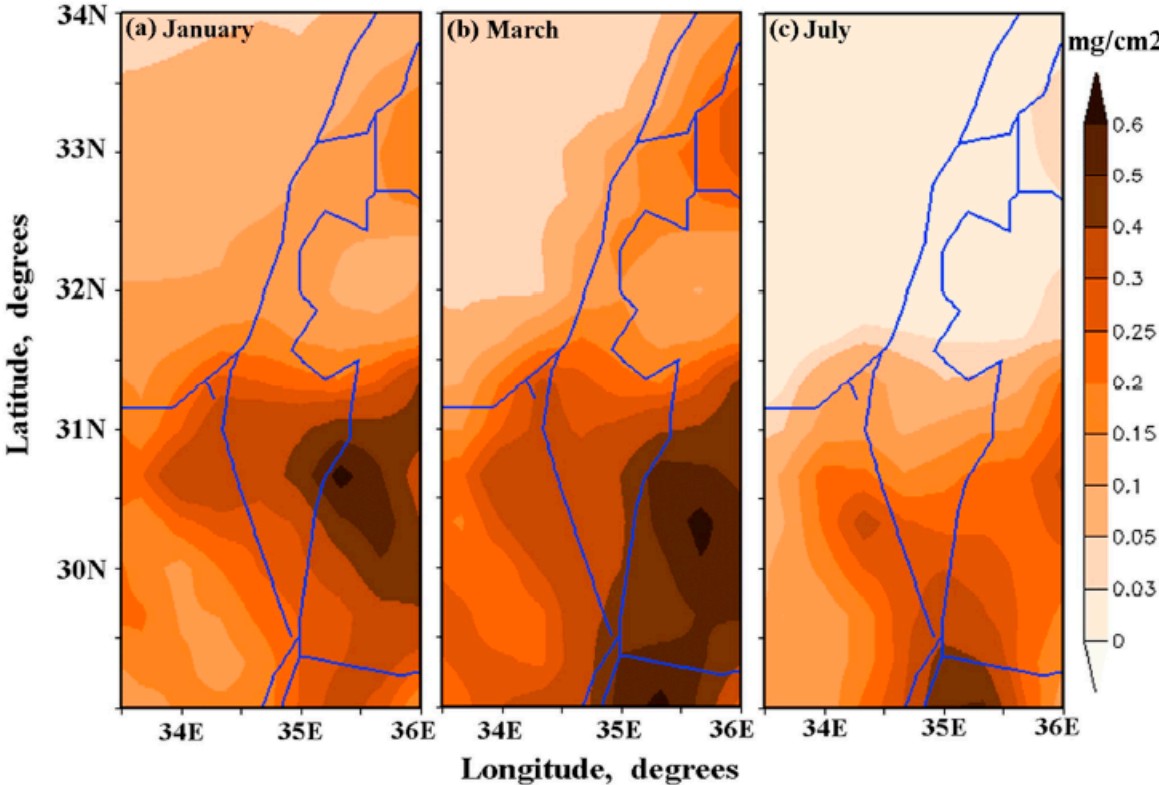

**Figure 12.** Maps of 14-year mean monthly-accumulated dust dry deposition in (**a**) January, (**b**). March, and (**c**) July.

*3.4. Trends*

To obtain long-term trends of DDD over the southern, central, and northern parts of Israel, the obtained monthly-accumulated model data at the specified sites were deseasonalized by removing 14-year averages (2006–2019) from any given month (Figure 13). The slope of a linear fit was used to determine trends of dust dry deposition (Figure 13). We found that there were no statistically significant trends (at the 95% confidence level) in dust dry deposition at any of the specified sites in Israel, during the 14-year study period: all the obtained *p*-values were essentially higher than 0.05 (Figure 13).

It was important to compare dust dry deposition trends in two different months, such as March and November. In March, pronounced spatial non-uniformity of DDD was observed (Figure 12b), whereas in November, DDD was almost uniformly distributed (Figure 11c). Figure 14 represents DDD trends in March and in November at the specified sites located in the southern, central, and northern parts of Israel. One can see that, in March, there were no statistically significant trends in DDD during the study period at any of the specified sites (Figure 14, the left panel). By contrast, in November, there were statistically significant trends in DDD at 95% confidence level in Haifa, Tel-Aviv, and Beer-Sheba: DDD increased from year-to-year during the 14-year study period (Figure 14, the right panel). This fact is illustrated by Figure 15, representing patterns of monthly-accumulated dust dry deposition in November in the first four years (2006, 2007, 2008, 2009) and in 2013, 2014, 2015, 2016. It is seen that DDD over northern Israel increased from year-to-year in November during the study period.

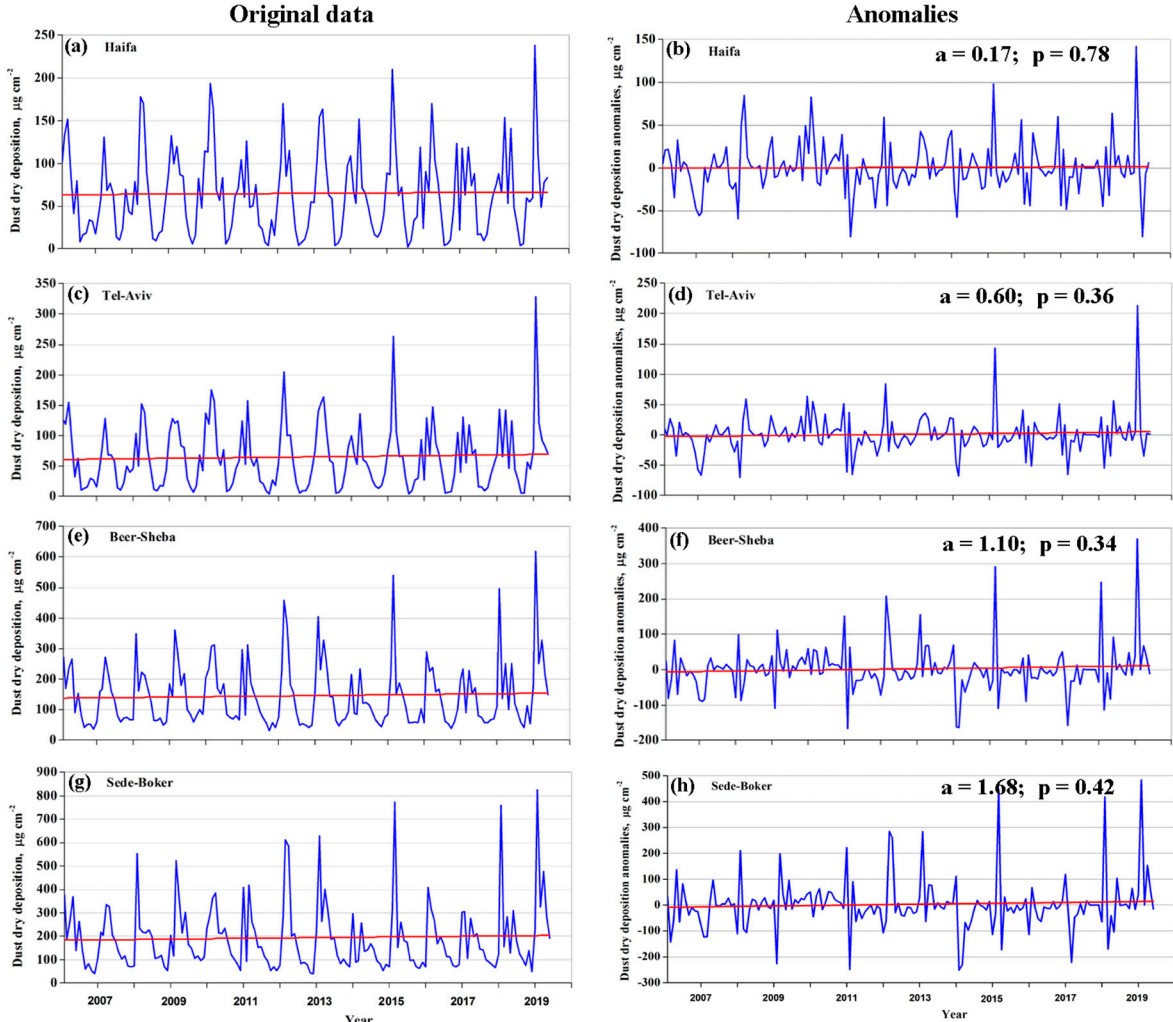

**Figure 13.** Month-to-month variations of DREAM monthly-accumulated dust dry deposition at the specified sites located in the southern, central, and northern parts of Israel: (a) and (b) – Haifa; (c) and (d) – Tel-Aviv; (e) and (f) – Beer-Sheba; (g) and (h) – Sede-Boker. The left column represents original monthly-accumulated dust dry deposition at the specified sites, while the right column represents their associated deseasonalized monthly anomalies. The straight red lines designate linear fits characterized by the slope a [µg cm$^{-2}$ year$^{-1}$] and $p$ values. One can see that all $p$ values were essentially higher than 0.05. Consequently, there were no statistically significant trends (at the 95% confidence level) in dust dry deposition at any of the specified sites during the study period.

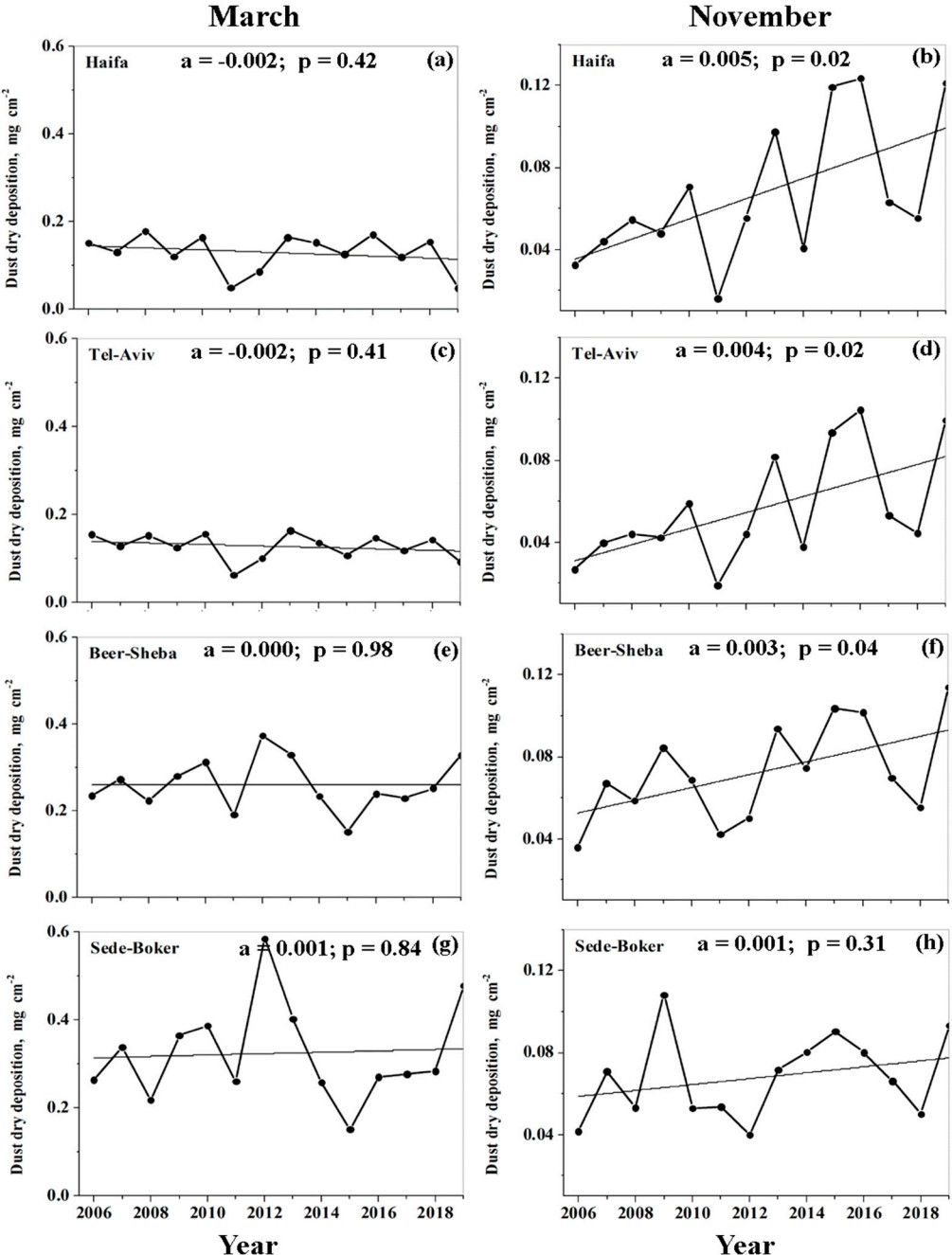

**Figure 14.** Year-to-year variations of DREAM monthly-accumulated dust dry deposition at the specified sites located in the southern, central, and northern parts of Israel in March (the left panel) and November (the right panel): (a) and (b) – Haifa; (c) and (d) – Tel-Aviv; (e) and (f) – Beer-Sheba; (g) and (h) – Sede-Boker. The straight lines designate linear fits, characterized by the slope a [mg cm$^{-2}$ year$^{-1}$] and $p$ values. In November, the $p$-values lower than 0.05 designate statistically significant trends in DDD at the 95% confidence level in Haifa, Tel-Aviv, and Beer-Sheba.

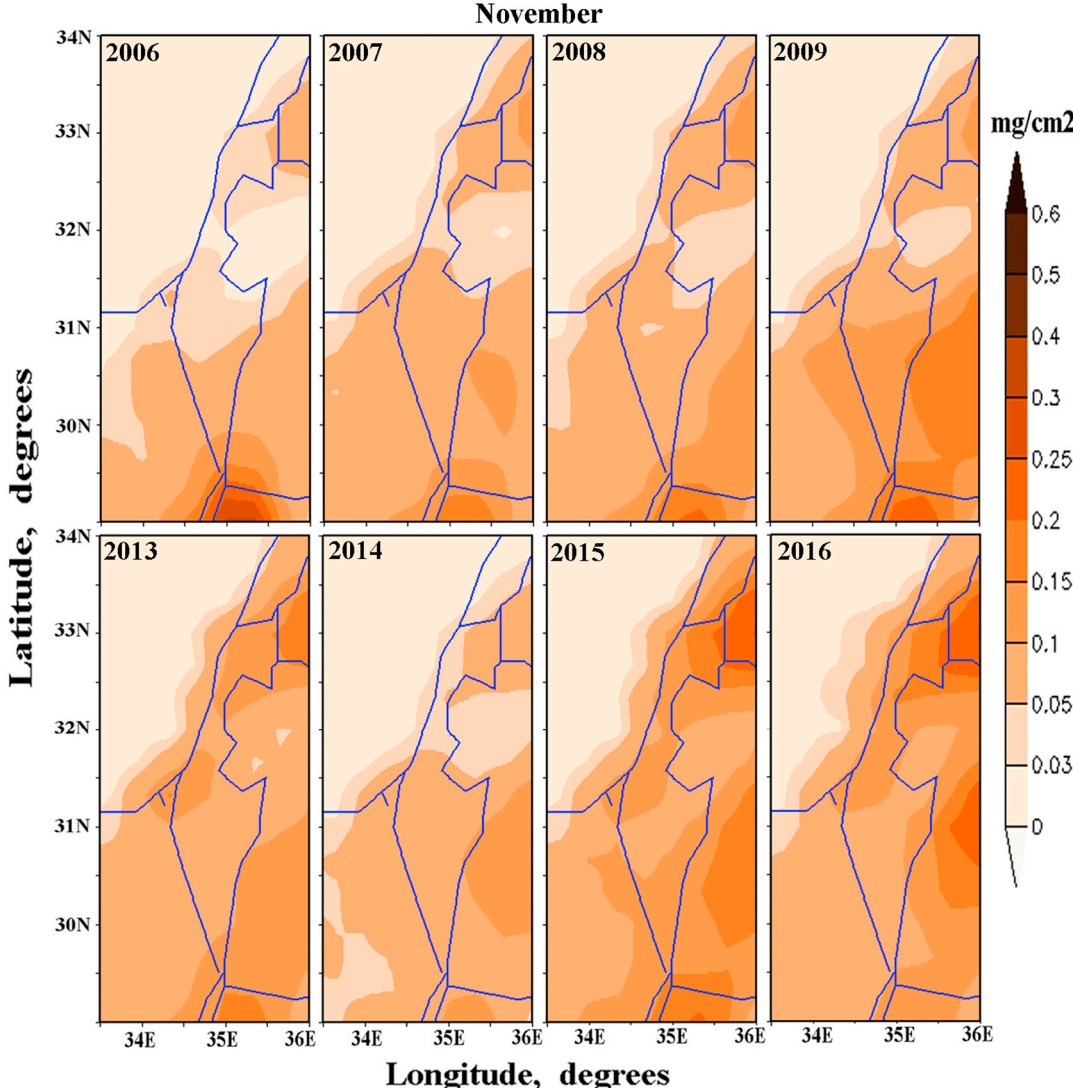

**Figure 15.** Comparison between monthly-accumulated dust dry deposition over the study region in November in 2006, 2007, 2008, 2009 (top panel) and that in 2013, 2014, 2015, 2016 (bottom panel).

We used a correlation matrix to investigate a relationship between DDD variations from year-to-year at the specified sites (Tables 2 and 3). One can see that, in November, the year-to-year variations of DDD in Haifa, Tel-Aviv, and Beer-Sheba were highly correlated due to increasing DDD (Table 2). In March, however, there was no correlation between these year-to-year variations of DDD in Haifa, Tel-Aviv, and Beer-Sheba (Table 3). Because of the proximity of Sede-Boker and Beer-Sheba to the Negev Desert, there is always a high correlation of 0.8 between DDD in those sites (Tables 2 and 3).

**Table 2.** Correlation matrix for year-to-year variations in DDD in Haifa, Tel-Aviv, Beer-Sheba, and Sede-Boker in November during the study period (2006–2019). The numbers represent correlation coefficients together with their standard error. The bold numbers highlight the high correlation between Beer-Sheba and Haifa, as well as between Beer-Sheba and Tel-Aviv in November.

|  | **Haifa** | **Tel-Aviv** | **Beer-Sheba** | **Sede-Boker** |
|---|---|---|---|---|
| Haifa | 1.00 | 0.99 ± 0.01 | **0.85 ± 0.10** | 0.40 ± 0.20 |
| Tel-Aviv | 0.99 ± 0.01 | 1.00 | **0.90 ± 0.10** | 0.40 ± 0.20 |
| Beer-Sheba | 0.85 ± 0.10 | 0.90 ± 0.10 | 1.00 | 0.80 ± 0.10 |
| Sede-Boker | 0.40 ± 0.20 | 0.40 ± 0.20 | 0.80 ± 0.10 | 1.00 |

**Table 3.** Correlation matrix for year-to-year variations in DDD in Haifa, Tel-Aviv, Beer-Sheba, and Sede-Boker in March during the study period (2006–2019). The numbers represent correlation coefficients together with their standard error. The bold numbers highlight the low correlation between Beer-Sheba and Haifa, as well as between Beer-Sheba and Tel-Aviv in March, compared to those in November (Table 2).

|  | **Haifa** | **Tel-Aviv** | **Beer-Sheba** | **Sede-Boker** |
|---|---|---|---|---|
| Haifa | 1.00 | 0.95 ± 0.03 | **0.10 ± 0.20** | −0.20 ± 0.20 |
| Tel-Aviv | 0.95 ± 0.03 | 1.00 | **0.30 ± 0.20** | −0.01 ± 0.20 |
| Beer-Sheba | 0.10 ± 0.20 | 0.30 ± 0.20 | 1.00 | 0.95 ± 0.03 |
| Sede-Boker | −0.20 ± 0.20 | −0.01 ± 0.20 | 0.95 ± 0.03 | 1.00 |

## 4. Conclusions

Similar quasiperiodic year-to-year variations of DDD with a two–three-year period were found over Israel and north-east Africa. The similarity indicates that the quasiperiodic variations were caused by the same factors. This phenomenon of quasiperiodic interannual variations of DDD has not been discussed in previous publications. The quasiperiodic interannual variations of DDD were associated with similar interannual variations of surface wind speed over Israel and north-east Africa. DREAM also showed similar seasonal variations of DDD over both Israel and north-east Africa, characterized by significant dust deposition in spring and a decrease in DDD from spring to autumn. These seasonal variations of DDD were associated with the similar seasonal variations of surface wind speed over Israel and north-east Africa. The obtained similar interannual and seasonal variations of DDD and surface wind indicate a causal link between them. Dust occurrence in the south-east Mediterranean is associated with cyclonic activity all over the Mediterranean basin [4,14]. The Mediterranean cyclones determine the surface winds over the whole area of the south-east Mediterranean, including north-east Africa and Israel, causing similar interannual and seasonal variability of DDD.

Using daily data from DREAM model runs at Tel Aviv University from 2006 to 2019, we found that, on average, during the 14-year study period, in the winter, spring, and summer months, the spatial distribution of monthly-accumulated dust dry deposition over Israel was non-uniform with the maximum of DDD over southern Israel. This maximum was determined by the significant contribution of local dust from the Negev Desert (and adjacent deserts in Sinai and Jordan) to the total dust deposition over Israel.

In the autumn months (November), DREAM showed an increase in dust dry deposition over northern and central Israel from year-to-year, resulting in the almost uniform distribution of DDD over Israel. As there is no local dust over northern and central Israel, it can be assumed that the observed increase in dust dry deposition from year-to-year in November was associated with increasing dust penetration from Syria.

The obtained specific features of spatio-temporal distribution of DDD will advance the general understanding of the dust-related problems, such as negative effects of DDD on the performance of solar panels in the south-eastern Mediterranean and in Israel in particular. Moreover, this knowledge is helpful for preventing insulator flashover in the regional power electric networks.

**Author Contributions:** All co-authors equally contributed to the methodology and writing of the current research article: P.K., E.V., B.S., P.A., S.N.; DREAM model development: S.N.; DREAM runs: P.K. All authors have read and agreed to the published version of the manuscript.

**Funding:** This research received no external funding.

**Acknowledgments:** The authors acknowledge support from the EU COST Action CA16202 "International Network to Encourage the Use of Monitoring and Forecasting Dust Products (InDust)". We thank Slavko Petkovic for help with the DREAM model. We thank all reviewers for their helpful comments.

**Conflicts of Interest:** The authors declare no conflict of interest.

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
