# Peer review of "Dust Dry Deposition over Israel"

_atmosphere, doi:10.3390/atmos11020197_

Round 1
Reviewer 1 Report
Please, see attach file.

Author Response
See our reply in the attached file.

Reviewer 2 Report
The caption for Figure 10 is present, but the figure is missing.
line 363 - northern
line 370 - northern
Author Response
See our reply in the attached file.

Reviewer 3 Report
This paper presents results about the comparison of year-to-year and seasonal variations of dust dry deposition between north-east Africa and Israel. The authors aim to investigate long-term trends in dust dry deposition during a period of 14 years. However, this manuscript should be subjected to a revision before acceptance:
L31 those previous findings should be highlighted on the document
L39 the scientific contribution should be clarify on the document, the new information obtained is not enough
L40 the dust-related problems in the southeastern Mediterranean and Israel should be mentioned on the document
L 51-56 In Introduction instead of Methodology
L86 the results of this comparison should be provided for each correlation
L103 A brief description of NASA MERRA aerosol reanalysis should be provided on Methodology section
L172 the correlation coefficient should be provided for each set of variables
L180 trajectory analysis should also be included on results
L183 please, provide probability of correlation
L228 trajectory analysis should also be provided on results, it is nor clear the contribution from Negev Desert
L275 the local contribution to dust deposition should be showed by a trajectory analysis
L298 the contribution to dust deposition from Syria should be showed by a trajectory analysis
L315 please, define “surface wind”
Author Response
See our reply in the attached file.

Reviewer 4 Report
The authors present a paper on DDD over Israel. They defend the existence of a similar quasiperiodic year-to-year variations of DDD over Israel and northeast Africa which may caused by the same factors. The study was carried out using daily Dust REgional Atmospheric Model (DREAM) runs at Tel Aviv University from 2006 to the present.
This is relevant scientific work, although it could be greatly improved as it has important shortcomings, in particular:
Improve the bibliographic review. There are many papers in this area that should be presented and discussed, taking the opportunity to demonstrate the originality and importance of your work; Although the model has already been validated in the past in other works, validation in new applications, as is the case, is good practice; The validation presented is poor, even with the argument that there is little data observed; it is suggested the use of tools available for this purpose such as the BOOT statistical package or the DELTA software (https://ec.europa.eu/jrc/en/scientific-tool/fairmode-delta-tool); (please see also https://doi.org/10.3390/fluids3010020); on the other hand, the validation should not only focus on particle deposition, but also on meteorological variables, at least those most relevant to the process, such as wind speed and direction; It is suggested that the issue of atmospheric circulations in the region be properly represented by one or several figures for the different seasons of the year - figures with a wind fields (pair wind direction and speed) estimated by the model; this type of approach would greatly facilitate the discussion and understanding of the phenomena associated with DDD in the region; It is suggested a more in-depth discussion of local effects, namely the importance of mesoscale circulations in the whole coastal area, since clearly deposition is conditioned by breeze circulations in those places.In my opinion, a weak validation of the results, removes robustness to the conclusions. On the other hand, greater importance should have been given to breeze movements in the coastal zone. Even if it is not within the scope of the work, its importance in DDD should be recognized.
Author Response
See our reply in the attached file.

Reviewer 5 Report
The authors show an important issue, i.e. the dust dry depositions. The methodology is corrected and the information reported are interesting. The section 2.1 "The DREAM model" should be reduced, the DREAM model is well known as well as fig 2 could be moved to the Suppler Mat. The main bias regards the introduction, it is really short and not informative. The authors must increase the introduction as well as the literature, it is really scarce. The dry deposition has an important role in atmospheric chemistry, the authors should cite eventual legislation on it, also by worldwide.
Author Response
See our reply in the attached file.

Round 2
Reviewer 1 Report
See attached file.

Reviewer 4 Report
The authors have done a good job and the manuscript has improved significantly although, in my opinion, validation remains the weakest point. Other statistical indicators could have been applied in addition to what was used. Correlation is not the only statistical indicator of model performance and it is for this reason that in the evaluation of models, regardless of the size of their domain or grid, several indicators are used that complement each other. It is true that the BOOT is widely used in local scale dispersion models, however it has several statistical indicators that could be useful also in your work.
Regarding the usefulness of DELTA Tools, with the necessary adaptations, it is perfectly suited to validate the DREAM model. DELTA tools has been extensively applied to regional models and clearly states in its user manual that its application "In theory the software works therefore independently of model gridding and spatial scale" (https://aqm.jrc.ec.europa.eu/public/data/DELTA_UserGuide.pdf).
The authors can find DELTA Tools applications for several regional chemical transport models such as CHIMERE or CAMX as well as meteorological ones such as Weather Research and Forecasting model (WRF).
A final note regarding mesoscale circulations. It is clear that DREAM is not suitable for the simulation of mesoscale circulations, but this does not mean that they do not exist and probably do not play a very relevant role in DDD in coastal regions. For this reason, although breeze circulations are outside the scope of your work, in my opinion, it would be relevant to draw attention to this weakness of your simulations in your paper.
